

# Mediterranean nekton traits: distribution, relationships and significance for marine ecology monitoring and management

Evangelos Tzanatos, Catherine Moukas and Martha Koutsidi

Department of Biology, University of Patras, Patras, Greece

## ABSTRACT

Biological traits are increasingly used in order to study aspects of ecology as they are related to the organisms' fitness. Here we analyze a dataset of 23 traits regarding the life cycle, distribution, ecology and behavior of 235 nektonic species of the Mediterranean Sea in order to evaluate the distribution of traits, identify rare ones, detect relationships between trait pairs and identify species functional groups. Trait relationships were tested using correlation and non-linear regression for continuous traits, parametric and non-parametric inference tests for pairs of continuous-categorical traits and cooccurrence testing for categorical traits. The findings have significant implications concerning the potential effects of climate change (e.g., through the relationships of the trait of optimal temperature), fisheries or habitat loss (from the relationships of traits related to tolerance ranges). Furthermore, some unexpected relationships are documented, like the inversely proportional relationship between longevity and age at maturity as a percentage of life span. Associations between functional traits show affinities derived from phylogenetic constraints or life strategies; however, relationships among functional and ecological traits can indicate the potential environmental filtering that acts on functional traits. In total, 18 functional groups were identified by Hill-Smith ordination and hierarchical clustering and were characterized by their dominant traits. For the assessment of the results, we first evaluate the importance of each trait at the level of population, community, ecosystem and landscape and then propose the traits that should be monitored for the regulation and resilience of ecosystem functioning and the management of the marine ecosystems.

# INTRODUCTION

Today, the characteristics of organisms, usually referred to as traits, are increasingly used in order to study aspects of biology and ecology as they are related to the organisms' ability to survive, grow and produce offspring, i.e., to influence their fitness (*Violle et al., 2007*). The so-called traits-based approaches, comprise an arsenal of methods that use traits instead of taxonomic information to characterize ecosystems, providing various advantages like alternative ways to study biological communities and their functioning (*Bellwood, Hoey & Choat, 2003*; *Bremner, 2008*; *Pecuchet et al., 2017*; *Pecuchet et al., 2018*). Furthermore,

Corresponding author
Evangelos Tzanatos,
tzanatos@upatras.gr

they can be used to elucidate the effects of environmental or anthropogenic stressors on ecosystem functioning (*Frelat et al., 2018*; *McLean, Mouillot & Auber, 2018*).

However, the study of biological traits is not novel in marine biology. Traits have long been studied in order to determine relationships between aspects of an organisms' physiology or ecology. Additionally, evolutionary processes have provided specific combinations of characteristics to organisms; hence, the traits appearing in any given species are not random (e.g., *Winemiller & Rose, 1992*; *Charnov, 1993*). It has been demonstrated that traits are not only related, as there are various relationships among them, but also there are trade-offs between them because of physiological constraints (*Litchman, Ohman & Kiørboe, 2013*). As the energy captured must be used to achieve maintenance, growth and reproduction of the organism, the allocation of energy between these in time (spanning aspects ranging from physiology to ecology) shapes the major aspects of the species' life histories. Trait combination into life histories serves functions like survival, growth, sexual maturation and reproduction at the organismal level (*Beverton & Holt, 1959*; *Beverton, 1992*) shaping fitness that can act at the level of populations and, ultimately, communities. However, the relationships between traits can be important, not only for understanding how evolution has shaped life into form and function (*Charnov, Gislason & Pope, 2013*), but also in order to assess the potential effects of environmental factors. Even more so as, apart from environmental filtering (*Bejarano et al., 2017*), anthropogenic stressors like fisheries can favour the selection of specific traits (e.g., *De Juan, Thrush & Demestre, 2007*) and thus indirectly alter future community trait composition, possibly modifying the functioning of communities or the ecosystem. As a result, there is a probability that the selection of specific traits may lead to changes in the frequencies of others that are associated with them as a side effect. This supposition gives novel importance to the study of relationships between traits. Furthermore, the study of trait frequencies and the identification of rare traits may indicate keystone species for ecosystem functioning (*Violle et al., 2017*).

In order to comprehend the functional aspect of traits and the implications of their relationships it is important to document their significance not only for shaping the life history of an organism at the level of the individual. Highlighting the importance of traits at the population scale comes easily, as traits are related to the adaptations of the species to its environment aiming to maximize fitness at the individual and population level (*Violle et al., 2007*). Various works have examined the significance of traits for the population/species possessing them (e.g., *Tornroos & Bonsdorff, 2012*; *Costello et al., 2015*; *Henseler et al., 2019*). *Villéger et al. (2017)* expand the relation of functioning at the population level to ecosystem processes and services. However, an overall evaluation of the importance or the implications of biological traits should be carried out at levels beyond that of the population, especially for nekton, where a multitude of species are under exploitation and inter-specific interactions have raised important challenges. Thus, a brief presentation of the significance and implications of nekton traits at various ecological levels, i.e., (a) that of the population, (b) interspecific relationships and communities, (c) overall ecosystem functioning and even (d) relevance for anthropogenic effects like fisheries and climate change is presented in Table 1.

Tzanatos et al. (2020), *PeerJ*, DOI 10.7717/peerj.8494

**Table 1  Significance of traits at the level of species, community, ecosystem functioning and anthropogenic effects.**

| a/a | Trait (example) | Significance or implications at the level of: | | | |
|-----|-----------------|------------------|-----------|-----------|----------------------|
| | | Population/species | Community | Ecosystem | Anthropogenic effects |
| 1 | Longevity (5 years) | Longer lifespan increases reproductive success over time (*Beauchard et al., 2017*)<br><br>May indicate population stability over time and potential of the various life stages to disperse (*Costello et al., 2015*) | Higher longevity renders individual more important both as prey and as a predator as more instances of predation | Longevity is related with natural mortality and thus with energy transfer in the ecosystem (*Charnov, Gislason & Pope, 2013*) | Longevity and age at maturity are related with the ability to recover from anthropogenic disturbance (*Kaiser et al., 2006*; *Rijnsdorp et al., 2016*) |
| 2 | Age-at-maturity (30% of lifespan) | Early maturity may increase resilience in unfavourable environmental conditions (*Bamber, 1995*)<br><br>Associated with cessation of growth (*Jonsson & Jonsson, 1993*) | NR | Ecosystem characteristics (e.g., productivity) may enhance or delay maturation | Longevity and age at maturity are related with the ability to recover from anthropogenic disturbance (*Rijnsdorp et al., 2016*)<br><br>Early maturation may increase resilience in high exploitation rates. Maturity significant for fisheries management (measures planned to ensure population part achieves sexual maturity) |
| 3 | Fecundity (5–10 eggs) | If low should ensure offspring survival-population fitness, as energy allocated to survival of offspring or fecundity (r/K-selection strategy) (*Pianka, 1970*) | High fecundity means higher abundance of young "defenceless" stages (eggs, larvae) that are possible prey for other populations, but higher inter-specific competition later on (*Bamber, 1995*) | As it provides easy-to-capture and rich in energy prey (compared to adult prey) may influence energy flow rates | Together with mortality until recruitment may affect stock size which is very relevant for fisheries (*Jennings, Kaiser & Reynolds, 2001*) |
| 4 | Hermaphroditism (gonochoristic) | Sexual maturity of the second (in succession) sex must be achieved through survival to guarantee successful spawning and recruitment<br><br>Size is important in determining male reproductive success (*Wooton, 1999*) | NR | NR | As both sex ratio and gear selectivity change with size, exploitation of one size part of the population may affect sex ratio and possibly reproductive success. |

Tzanatos et al. (2020), *PeerJ*, DOI 10.7717/peerj.8494

**Table 1** (*continued*)

| a/a | Trait (example) | Significance or implications at the level of: | | | |
|-----|-----------------|-----------------|-----------|-----------|----------------------|
| | | **Population/species** | **Community** | **Ecosystem** | **Anthropogenic effects** |
| 5 | Maximum length (20 cm) | Related to individual biomass, food web position, abundance, metabolic rates, and dispersal (*Costello et al., 2015*) | As in the marine ecosystem there is an "eat what is smaller" pattern, there has to be some variability in species sizes to support a community (*Giacomini, Shuter & Baum, 2016*; *Kerr & Dickie, 2001*) | Related to energy flow in the ecosystem (because of association with trophic level/diet) and resulting food webs (*Gerlach, Hahn & Schrage, 1985*; *Jennings, 2005*) | Relevant for fisheries (with regard to body shape) for selectivity (*Jennings, Kaiser & Reynolds, 2001*) |
| 6 | Body form (flat) | Related to position in the water column/habitat, diet/potential prey, activity (*Henseler et al., 2019*; *Wooton, 1999*) | Because of association with habitat, specific communities may have higher frequencies of some body forms | NR | Related to the way fishing gear may affect selectivity (together with size) (*Jennings, Kaiser & Reynolds, 2001*) |
| 7 | Optimal depth (0–50 m) | Physical factor determining potential species habitat (*Costello et al., 2015*) | Depth is a major factor shaping marine communities (*Pereira et al., 2018*; *Vega-Cendejas & De Santillana, 2019*) | Depth may affect productivity and energy flow as e.g., below the euphotic zone the lack of primary production modifies trophic links. Also effects of elements like critical depth or seagrass bed distribution (*Kaiser et al., 2006*) | Different gears and fishing sectors are often operating in different depths resulting in different communities prone to exploitation and resulting catch composition (*Tserpes, Tzanatos & Peristeraki, 2011*) |
| 8 | Optimal temperature (25–30 °C) | Defines optimal temperature conditions for population fitness. May affect movement between water masses (behavioural thermoregulation) and thus abundance and distribution (*Moyle & Cech Jr, 2004*) | Due to climate change can shift to dominance of more thermophilic species (*Lejeusne et al., 2010*) | NR | More thermophilic species may appear more frequently in the catches ((*Cheung, Watson & Pauly, 2013*); *Vasilakopoulos et al., 2017*) |
| 9 | Habitat type (benthic) | Populations are closely associated to pelagic or benthic habitat or migrate between them (*Henseler et al., 2019*; *Wooton, 1999*) | Specific habitats are characterised by specific communities (*Ballesteros, 2006*; *Kalogirou et al., 2010*) | Effect of seabed type on ecosystem functioning expected to be significant as both are related to biodiversity and its attributes (*Loreau et al., 2001*) | Has implications for target species abundance and bycatch (thus fishing gear use) (*Tzanatos et al., 2006*) |

Tzanatos et al. (2020), *PeerJ*, DOI 10.7717/peerj.8494

**Table 1** (*continued*)

| a/a | Trait (example) | Population/species | Community | Ecosystem | Anthropogenic effects |
|-----|-----------------|--------------------|-----------|-----------|------------------------|
| | | | | **Significance or implications at the level of:** | |
| 10 | Distribution (tropical) | Related to proximity to the geographic distribution of the area examined (if e.g., through Gibraltar or Suez in the Med) and could be associated to favourable environmental conditions (*Coll et al., 2010*) | Species of alien distribution (invasive) species may dominate the community through colonisation of empty niches/lack of "natural enemies" (*Givan et al., 2017*) | NR | Climate change or other environmental changes may be forcing changes in distribution paterns (*Galil, 2007*, *Occhipinti-Ambrogi & Savini, 2003*) |
| 11 | Sea bed type (hard) | Physical factor determining potential species habitat (*Costello et al., 2015*) | Specific habitats host and are characterized by specific communities (*Ballesteros, 2006*; *Kalogirou et al., 2010*) | Effect of seabed type on ecosystem functioning expected to be significant as both are related to biodiversity and its attributes (*Loreau et al., 2001*; *Solan, Aspden & Paterson, 2012*) | Has implications for target species abundance and bycatch (thus gear use) (*Tzanatos et al., 2006*) |
| 12 | Spawning habitat (pelagic) | Spawning habitat determines the nature and intensity of hazards encountered by eggs and larvae (*Leis, 2006*; *Wooton, 1999*) | May determine seasonal communities as a result of spawning seasonality and also of populations feeding on eggs and juveniles | If spawning habitat different from adult stage habitat may be relevant to benthopelagic coupling (*Leis, 2006*; *Secor, 2015*) | May create aggregations prone to fisheries (*Erisman et al., 2017*) |
| 13 | Temperature range (eurythermal) | May increase population resilience to abrupt temperature changes or ability to change environment | Eurythermal species may dominate community under climate change/frequent weather changes | NR | May increase population resilience to climate change, invasion rates and appearance in the fisheries catch<br><br>Eurythermal species may be favoured in thermal pollution sites (*Bamber, 1995*) |
| 14 | Salinity range (stenohaline) | May be related to population ability to approach/enter productive habitats like estuaries & lagoons (*Moyle & Cech Jr, 2004*) | Shapes communities of brackish waters, e.g., along salinity gradients (*Henriques et al., 2017*; *McLusky & Elliott, 2007*; *Pasquaud et al., 2015*) | Relevant to matter & energy transfer between the ocean and brackish waters through euryhaline species (*Martino & Able, 2003*) | Shapes the resources exploited by fisheries in brackish environments (e.g., lagoon fisheries) (*Katselis et al., 2003*) |

**Table 1** (*continued*)

| a/a | Trait (example) | Significance or implications at the level of: | | | |
| --- | --- | --- | --- | --- | --- |
| | | **Population/species** | **Community** | **Ecosystem** | **Anthropogenic effects** |
| 15 | Depth range (eurybathic) | Eurybathic species have more potential habitat and might be more resilient to habitat loss (*Costello et al., 2015*) | Communities dominated by eurybathic species may be more resilient to environmental changes and disturbance | Eurybathic species may transfer energy through depth zones and contribute to benthopelagic coupling | More eurybathic species may be more resilient to habitat degradation by fisheries or other anthropogenic effects |
| 16 | Seasonal migrations (migratory) | Can change the population ecological status, may lead to a seasonal (periodic) life strategy and shape seasonal energy needs (*Secor, 2015*; *Winemiller & Rose, 1992*) | Community will change seasonally, qualitatively and quantitatively (*Park et al., 2019*) | Can have impact on energy flow, creating seasonal dynamics (*Secor, 2015*) | Many fisheries are based on seasonal migrations for fishing grounds or even operation of specific gears (e.g., lagoon fisheries) (*Katselis et al., 2003*) |
| 17 | Trophic level (3.5–4.2) | Derived from the type and frequency of trophic objects in its diet (*Costello et al., 2015*) | Influence on other species abundance and community structure and dynamics (*Costello et al., 2015*) | May alter nutrient cycling in the ecosystem (*Beauchard et al., 2017*) | Depending on exploitation removing part of the trophic network may result in fishing down the food web (*Pauly et al., 1998*) |
| 18 | Diet (zooplankton) | Determines food web position (*Costello et al., 2015*) | Influence on other species abundance and community structure and dynamics (*Costello et al., 2015*) | May alter nutrient cycling in the ecosystem (*Beauchard et al., 2017*) | Relevant to fishing gear mode of operation exploiting diet (hook and line gears e.g., longlines) and associated target species & catch composition |
| 19 | Spawning period (spring) | Shapes the period that the population must feed to prepare spawning and non-feeding period. May be associated with "weak" period (bad condition) after spawning (*Dutil, 1986*; *Engelhard & Heino, 2005*) | May shape feeding interactions and trophic links within the community seasonally, both as a result of preying on eggs and larvae and, secondarily, because of the seasonal pattern of recruitment (*Edworthy & Strydom, 2016*) | Is affected by suitability of the environmental conditions for eggs & larvae. Is affected by energy supply (low energy may result in delay or skipping spawning). As the spawning period generates eggs and larvae it provides potential prey (*Rideout, Morgan & Lilly, 2006*; *Wooton, 1999*) | Seasonality of fisheries may lead to unsuccessful spawning and result in few individuals recruited |
| 20 | Feeding type (plankton) | Related to the diet and the trophic level through the relative size and mobility of the prey in comparison to the predator (*Costello et al., 2015*) | By shaping diet can affect the community composition | Related to prey community composition and lower trophic level succession patterns (*Mariani et al., 2013*) | Relevant to fishing gear mode of operation exploiting feeding behaviour (hook and line gears e.g., trolling lines, longlines) and associated target species & catch composition (*Jennings, Kaiser & Reynolds, 2001*) |

Tzanatos et al. (2020), *PeerJ*, DOI 10.7717/peerj.8494

**Table 1** (*continued*)

| | | Significance or implications at the level of: | | | |
|---|---|---|---|---|---|
| a/a | Trait (example) | Population/species | Community | Ecosystem | Anthropogenic effects |
| 21 | Sociability (schools) | Benefits like predation avoidance, food location and foraging strategy, improvement of reproductive success (*Pitcher & Parrish, 1993*; *Wooton, 1999*; *Krause & Ruxton, 2002*)<br><br>Costs like competition for food or mate, predator attraction, disease transmission (*Côté & Poulin, 1995*; *Krause & Godin, 1995*) | Schooling important in shaping communities regarding hydrodynamic characteristics (*Floeter et al., 2007*) | Schooling/pelagic fish may colonise new habitats (e.g., reefs) more easily (*Paxton et al., 2018*) | Relevant to fishing gear mode of operation exploiting gregarious fish behaviour (e.g., purse seines) and associated target species & catch composition (*Jennings, Kaiser & Reynolds, 2001*)<br><br>Schooling/pelagic fish may colonise new artificial habitats (e.g., reefs) more easily (*Paxton et al., 2018*) |
| 22 | Exposure (cryptic-temporarily) | Population must balance ability to graze/predate and predation avoidance<br><br>Population (especially cryptic) may have developed diel activity rhythms (*Matheson et al., 2017*) | Depending on conditions (e.g., habitat type) cryptic species may dominate communities (*Schrandt et al., 2018*)<br><br>Level of exposure and cryptic behaviour relevant to differences in diel community composition (*Matheson et al., 2017*), | NR | NR |
| 23 | Mobility (high) | Indicates a dispersal potential and a more or less mobile lifestyle (*Costello et al., 2015*) | Might differentiate pelagic (more motile) from benthic (more static) communities | May be relevant to transfer of energy between ecosystems or benthopelagic coupling | Relevant to fishing gear mode of operation exploiting fish motility behaviour (e.g., nets) and associated target species & catch composition (*Ferno & Olsen, 1994*; *Jennings, Kaiser & Reynolds, 2001*) |

**Notes.**

NR, Not relevant. References with explanation/examples are indicated with numbers corresponding to in-text citations following and are listed in detail in the Reference list.

While the term "trait" generally refers to morphological, physiological or phenological features that are measurable at the individual level without reference to the environment, (*Violle et al., 2007*) define as functional traits the characteristics that may affect survival, growth, or reproduction and thus indirectly impact fitness. *Beauchard et al. (2017)* determine as ecological traits characteristics related to environmental preferences (like optimal temperature or depth). Often, analyses of trait patterns focus on functional traits (e.g., to determine how evolution has shaped life strategies), however the inclusion of data on ecological traits in the analyses can provide insights into how the environment may affect aspects of organismal fitness by associations with functional traits.

The investigation of patterns and relationships between traits are especially important for regions that host a high biodiversity like the Mediterranean Sea (*Myers et al., 2000*; *Bianchi & Morri, 2000*). Furthermore, the Mediterranean has a long history of human presence and is today facing challenges due to stressors that are either novel or are acting at unprecedented levels (*Lejeusne et al., 2010*). *Koutsidi et al. (2016)* have already studied traits relationships and rarity; however, in that work traits had been assembled only for a limited number of species (mainly commercial species targeted or caught by fisheries). Additionally, in that work traits had been evaluated only as categorical variables (i.e., with each trait being described by trait categories). This may possibly have blurred the results in the case of quantitative traits like life span or size.

The aim of the present study is to use an extensive dataset of 23 traits assembled from the bibliography for 235 nektonic species of the Mediterranean Sea in order to: (a) determine whether there are rare or dominant traits by evaluating the distribution of traits, (b) detect whether there are undocumented relationships between pairs of biological traits by looking for patterns between functional traits, ecological traits or both, (c) identify functional groups and (d) evaluate selected rare traits with regard to climate change and to species' resilience to human impact.

## MATERIALS & METHODS

Information on 23 traits related to the life cycle, distribution, ecology and behaviour of 235 species of nekton (217 fish, 10 cephalopods, 8 crustaceans) occurring in the Mediterranean Sea was collected from the bibliography (Table 2). For species selection, an already existing dataset assembled and analyzed in the work of *Koutsidi et al. (2016)* was extended, using as criteria to: (a) cover the catches of the common commercial fishing gears as much as possible. Compared against the onboard sampling catch composition data, i.e., species both landed and discarded by fishermen, from the application of the European Union Data Collection Framework (EC 199/2008) in the eastern Ionian Sea (GFCM area: GSA20) in 2013–2014 the species accounted for over 98% of the catches in terms of biomass (both total catches and also catches by gear, which for some gears reached 100%), (b) adequately depict the traits of the fisheries landings in a pan-Mediterranean scale. Compared to the FAO-GFCM landings dataset of 2015, after excluding general organismal categories (like "fishes" or "mollusks"), the dataset corresponded to 75% of the total landings (which naturally also include various benthic species like bivalves) by taxon, (c) include all the

Tzanatos et al. (2020), *PeerJ*, DOI 10.7717/peerj.8494

Peer J

**Table 2** **List of traits used in the analyses, trait type (functional/ecological), variable type and categories/modalities used for categorical traits.**

| Trait | Trait type | Variable type | Trait categories/modalities (in case of categorical trait) | | | | | |
|---|---|---|---|---|---|---|---|---|
| Longevity | Functional | CON | | | | | | |
| Age at maturity* | Functional | CON | | | | | | |
| Fecundity** | Functional | RAN | | | | | | |
| Gonochorism | Functional | CAT | Gonochoristic | Hermaphrodite | | | | |
| Maximum length | Functional | CON | | | | | | |
| Body shape | Functional | CAT | Flat | Long | Deep | Atractoid | Rounded | |
| Optimal depth | Ecological | RAN | | | | | | |
| Optimal temperature | Ecological | RAN | | | | | | |
| Habitat type | Ecological | CAT | Pelagic | Benthic | Benthopelagic | | | |
| Distribution | Ecological | CAT | Global | Temperate | Tropical | Subtropical | | |
| Seabed morphology | Ecological | CAT | Open sea | Soft | Hard | Variable | | |
| Spawning habitat | Ecological | CAT | Pelagic | Benthic | | | | |
| Temperature range | Ecological | CAT | Stenothermal | Eurythermal | | | | |
| Salinity range | Ecological | CAT | Stenohaline | Euryhaline | | | | |
| Depth range | Ecological | CAT | Eurybathic | Stenobathic | | | | |
| Seasonally migratory | Ecological | CAT | Migratory | Non-migratory | | | | |
| Trophic level | Functional | RAN | | | | | | |
| Diet | Functional | CAT | Herbivore | Zoobenthivore | Zoobenthivore-Hyperbenthos | Omnivore | Zooplankton | Piscivore |
| Spawning period | Functional | CAT | Winter | Spring | Summer | Autumn | All year | |
| Feeding behaviour*** | Functional | CAT | Grazer*** | Active predator | Ambushing predator | | | |
| Sociability | Functional | CAT | Schools | Shoals-large groups (>10) | Small groups (<~10) | Solitary | | |
| Exposure | Functional | CAT | Free | Cryptic (permanently) | Cryptic (temporarily) | | | |
| Mobility | Functional | CAT | Ambusher | Small | Medium | High | | |

**Notes.**
CON, Continuous; RAN, Continuous, provided as a range for most/all species; CAT, Categorical; *, as % of maximum age; **, scale of eggs/juveniles per spawn, maximum value indicated; ***, indicating that food items have negligible or low mobility related to predator.

species found in highly diverse habitats: For this we included the inventory of species from Posidonia beds and coralligène formations, that comprise the two marine habitats with the highest biodiversity in the Mediterranean. Regarding coralligène formations, information was obtained from the review article by *Ballesteros (2006)*. As there was no review on the nekton of Posidonia beds, we included the lists from five different publications -that use different sampling techniques and come from different areas of the Mediterranean Sea: *Francour (1997)*, *Guidetti (2000)*, *Fernandez et al. (2005)*, *Moranta et al. (2006)*, *Kalogirou et al. (2010)*, (d) Incorporate species distributed in lagoons that are highly productive ecosystems with brackish characteristics (*Nicolaidou et al., 2005*). Overall, we included 68 species from Posidonia beds, 26 species from coralligène formations and 28 species from lagoons, (e) finally, 22 Lessepsian species (species invading the Mediterranean Sea through the Suez canal), as listed by the review of *Corsini-Foka & Economidis (2007)*, were also included. Even though some of these 235 species can be characterized as benthic (e.g., *Octopus vulgaris*, *Nephrops norvegicus* or fish species living in burrows/crevices) they were included in the dataset as they mainly interact with nekton and are not sessile.

Trait information is often found in printed sources (e.g., fish identification keys) and terms are not standardized. Thus, no systematic review could be performed. Instead, the information was collected from books, review articles or journal research articles by searching using the species name and adding the name of the trait or relevant terms (e.g., "life span" instead of "longevity", "reproduction" instead of "spawning period"). Regarding reference sources, we preferred to use information from peer-reviewed publications or books over grey literature. As the objective was to collect data on Mediterranean species, when there was a unique reference about a species it was used regardless of the source, however, if there was information coming from locations outside and inside the Mediterranean (e.g., from two different papers), preference was given to the latter. If there were multiple sources of information from different areas within the Mediterranean (e.g., from various papers), we chose the references from the central Mediterranean (Ionian Sea). Regarding habitats, again we preferred information coming from marine habitats (e.g., rather than brackish waters or lagoons) if a species is distributed in many habitat types. In the case of species that can be found over different substrates and we had different publications with variable trait values we tried to focus on the most common habitat.

The various traits comprise different types of variables: continuous (e.g., size or maximum lifespan), range (e.g., depth) and categorical (e.g., spawning habitat: pelagic or benthic). Concerning categorical traits, each species was assigned to one trait category/modality per trait. The definition of these 23 traits and their modalities is provided in Table S1. The information on traits as continuous variables and the information on traits categories per species as well as the bibliographic reference for the documentation of each trait per species can be found at: https://figshare.com/articles/Koutsidi_Moukas_Tzanatos_23_biological_traits_of_235_species/11347406.

For the detection of patterns in their distribution, the range type traits (fecundity, optimal depth, optimal temperature, trophic level) were expressed as such, while for statistical tests and the detection of relationships between trait pairs they were used as the

average between the minimum and the maximum (thus used as continuous type traits). Continuous and range-type traits were investigated for outlier detection using the Grubbs test (*Grubbs, 1950*). As this test is parametric, it was performed after the log-transformation of the traits: longevity and maximum length, while age at maturity was transformed using the square root to achieve normality. As average fecundity and average depth could not be described by a normal distribution with any possible transformation the Grubbs test was not used on these variables. Regarding the identification of rare traits, the distribution of trait values (for continuous variable-type trait) or the frequencies of trait categories (for categorical variable-type trait) allowed the identification of rare traits as these shared by less than 5% of species.

To identify potential relationships the continuous traits longevity, fecundity, maximum length and depth were transformed using the natural logarithm. Each of the total of 23 traits was examined for the existence of a potential relationship with all other ones, depending on their types. Regarding pairs of continuous traits, Pearson correlation between all pairs of continuous traits was used. As carrying out a test for multiple hypotheses increases the probability of a rare event, the likelihood of incorrectly rejecting the null hypothesis (Type I error) increases; hence in the results the Bonferroni correction was incorporated. Correlation has the advantage of investigating relationships without assuming causality, however it can only detect linear relationships. In cases where the residuals indicated a non-linear pattern, a polynomial regression was additionally used to investigate the existence of non-linear relationships and, in cases of better fit, these relationships are presented instead.

For the detection of relationships between continuous and categorical traits, the $t$-test or Analysis of Variance (ANOVA) was used, depending on the number of categories of the categorical trait. As these tests are parametric, the Shapiro–Wilk's test was used to test for normality. Fischer's exact test (for traits including only two trait categories) or Levene's test (for traits with more than two trait categories) were used to examine the homogeneity of variances. Regarding normality, only longevity and maximum length (their logarithm) were found to follow a normal distribution. As in general, both ANOVA and $t$-test are considered as robust inferential tests (*Zar, 1999*), if the tests for homogeneity of variance did not reject the null hypothesis the parametric tests were used. In the cases where the parametric prerequisites were not fulfilled, the Mann–Whitney or Kruskal-Wallis tests were used instead (*Zar, 1999*). Similarly, we incorporated the Bonferroni correction here.

To detect relationships between categorical traits, we followed the approach by *Koutsidi et al. (2016)* that had been used for a smaller number of traits (21) and species (86). This approach investigates pairwise patterns of traits co-occurrence and compares these co-occurrences to the number of randomly expected ones, characterizing them thus as positive (the pair of traits tends to co-occur more than expected by the product of their independent frequencies), negative (the pair of traits tends to co-occur less than expected by the product of their independent frequencies) or random co-occurrences. For this analysis we used the library "co-occur" (*Griffith, Veech & Marsh, 2016*) in R-language (*R Core Team, 2019*).

For the identification of species functional groups, first we carried out a Hill-Smith ordination (*Hill & Smith, 1976*) of the species for which we had a complete 23 trait dataset. This type of analysis can handle both continuous (range-type traits were analyzed as continuous variables using their average) and categorical variables concurrently. In order to provide equal weights to all traits we expressed all continuous-type traits as percentages of their maximum value. From the results of the Hill-Smith ordination we analyzed the species coordinates on the 11 major axes derived (58% of variance explained) by hierarchical clustering carried out using Ward's method, similar to *Benedetti, Gasparini & Ayata (2016)*, *Benedetti et al. (2018)*. For this analysis we used the library "ade4" in R (*Dray & Dufour, 2007*).

The groups identified by the above method, were then characterized by their traits. Regarding categorical traits: (a) we considered that if more than 90% of the species of a group share a specific modality of a certain trait then the entire group is characterized by this modality (as a major percentage of species possess it). Furthermore, (b) we highlighted modalities that have particular importance for a group, i.e., (even if not shared by more than 90% of the species) were found in a percentage that was more than 20% (arbitrarily defined) higher compared to that modality percent frequency in the species pool. For continuous traits, we compared the averages of the 18 functional groups through ANOVA (optimal temperature) or their medians through Kruskal–Wallis (all other traits, as the parametric prerequisites were not fulfilled) and, in case of statistical significance, indicate the groups with highest/lowest averages or medians in the functional group characterization.

The significance level $\alpha = 0.05$ was used for all inferential tests, subsequently incorporating the Bonferroni correction as stated above.

# RESULTS

Trait frequencies and distributions allowed the identification of dominant and rare traits in the 235 species examined. Regarding continuous traits, 89.6% of the species examined were found to have a maximum life span of up to 25 years (minimum: 1.1 years, maximum: 118.0 years, average: 14.4 years, median: 9 years), but life cycles can reach up to ∼120 years with the rare high longevity frequency classes shared in most cases by only a few species (Fig. 1A). Concerning age at maturity as a percentage of lifespan (minimum: 2.0%, maximum: 75.0%, average: 25.2%, median: 21.8%), both early and late age at maturity come as outliers with low frequencies (Fig. 1B). Finally, regarding size (minimum: 5.0 cm, maximum: 800 cm, average: 63.7 cm, median: 37.4 cm), nekton species with a maximum length over ∼80cm were uncommon (∼78% of species examined may reach below this size), while nekton reaching 2-8 m were scarce. Slightly over 5% were the number of nekton species with a maximum size below 10 cm (Fig. 1C). No outliers were found in the log-transformed longevity (Grubbs test, $G = 2.82$, $p = 0.91$) and maximum length data (Grubbs test, $G = 3.25$, $p = 0.23$) or in the square root age at maturity (Grubbs test, $G = 3.44$, $p = 0.10$).

For range type traits (Figs. 1D–1G), fecundity showed a distribution where the extreme values (low and high fecundities) are represented by only a few species, whereas the majority
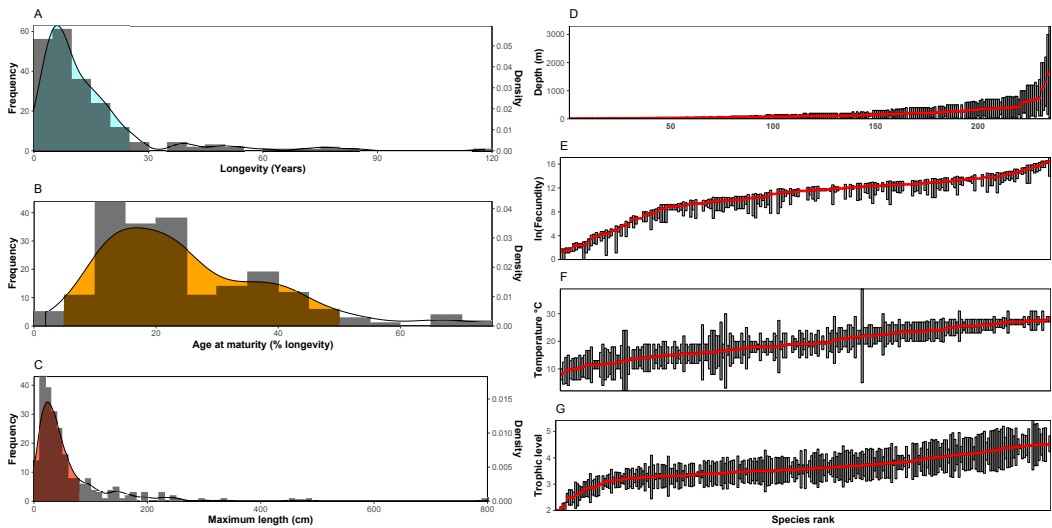

**Figure 1** (A–C) Distribution of continuous traits of the nekton species examined. (D–G) Ranges and/or means of range-type traits. Species are ranked according to the mean of the range. Note: In the bibliography, sometimes fecundity is provided as the maximum number of offspring with no indication of the minimum. In these cases it is here denoted not as a range, but with the same symbol as the mean.

of species demonstrate very similar fecundity ranges (also indicated by the S-shaped curve in the figure). Especially the values of low fecundity are mostly associated with Chondrichthyes (the 20 species with the lowest fecundity values are all cartilaginous fish). The highest values of fecundity were obtained by a heterogeneous group of species (*Thunnus thynnus, Labrus merula, Psetta maxima, Scophthalmus rhombus, Polyprion americanus*). Distribution across depth is relatively continuous with many species being distributed in shallow depths (40 species with an average depth shallower than 20 m and 86 above 50 m). Only a few species were found to be distributed in deep habitats (only 17 had an average depth deeper than 500 m, but the case is that deep-living nekton generally includes species whose traits are relatively unknown and, as a result, were generally not included in the dataset of the 235 species examined here). The distribution of trophic level indicated the rarity of low (reaching up to trophic level 3) trophic level species among the nekton, while the distribution of optimal temperatures did not indicate important outliers. The Grubbs test indicated no outliers regarding average trophic level (Grubbs test, $G = 3.14$, $p = 0.34$) and optimal temperature (Grubbs test, $G = 2.15$, $p = 0.99$).

Regarding categorical traits (Fig. 2), gonochorism (86%), subtropical distribution (64%), summer spawning (61%), free exposure type (67%), benthopelagic habitat use (63%), grazing feeding type (60%), eurythermal temperature range (66%), stenohaline salinity range (61%) and solitary behavior (57%) were dominant trait categories among the species examined. Flat (9%) and long body shape (7%), tropical (9%) and cosmopolitan geographic distribution (3%), autumn (3%), winter (7%) and all year spawning (6%), hard substrate seabed type (9%), ambusher mobility (6%), ambushing predation feeding

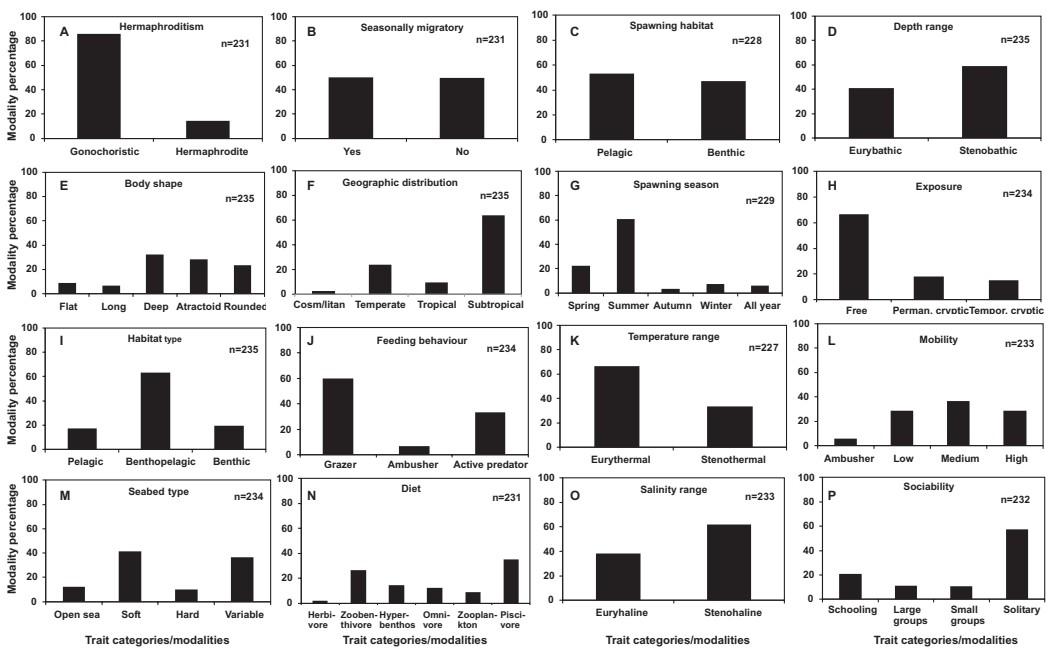

**Figure 2** **Frequency distribution of categorical traits of the nektonic species examined.** (A) Hermaphroditism, (B) Seasonal migrations, (C) Spawning habitat, (D) Depth range, (E) Body shape, (F) Geographic distribution, (G) Spawning season, (H) Exposure, (I) Habitat type, (J) Feeding behaviour, (K) Temperature range, (L) Mobility, (M) Seabed type, (N) Diet, (O) Salinity range, (P) Sociability.

behaviour (7%) and herbivorous (2%) and zooplankton diet (9%) were found to be shared by below 10% of the total species.

The relationships between continuous traits indicated eight statistically significant correlations, incorporating the Bonferroni correction (Table 3, Fig. 3). Longevity had most correlations, with three positive ones (long-living organisms are larger, have high trophic level and dwell deeper) and one negative (long-living organisms have lower age at maturity –as a percentage of life span). Fecundity was found to increase with optimal temperature (Fig. 3J), while maximum length increased with both trophic level and depth (Figs. 3B, 3E). Depth and trophic level were also positively correlated (Fig. 3H). Finally, in three cases, non-linear relationships provided better residual fit than linear ones (Figs. 3C, 3F, 3I): fecundity was higher in species of low and high longevity ($R^2 = 0.286$, $p < 0.001$), fecundity had the highest values in intermediate depths ($R^2 = 0.055$, $p = 0.004$) and optimal temperature was higher in species of intermediate depths ($R^2 = 0.046$, $p = 0.008$), in the last two cases the correlation model explaining a low percentage of variance.

The relationships between continuous and categorical traits indicated 20 cases where there are significant statistical differences in the value of a continuous trait between the different modalities incorporating the Bonferroni correction (Table 4). The main findings are summarized in Table 5 (but see Fig. S1 for pairwise comparisons between trait categories). Longevity was highest in flat-shaped species, in ambushing and active predators, piscivorous species and pelagic spawners. Fecundity was higher in atractoid

**Table 3  Pearson correlation coefficients between continuous traits.** Statistically significant correlations without taking into account the Bonferroni correction are denoted in red color, while these incorporating the Bonferroni correction are indicated in green. Pairs of traits where non-linear regression indicated that a non-linear relationship was better than a linear one in describing trait fluctuations are indicated by an asterisk (in all these cases the non- linear $p < 0.01$ remained significant after the Bonferroni correction).

| Trait | ln(Longevity) | Age at maturity | ln(Fecundity) | ln (Maximum length) | ln (Depth) | Trophic level | Optimal temperature |
|---|---|---|---|---|---|---|---|
| ln(Longevity) | – | | | | | | |
| Age at maturity | −0.34 | – | | | | | |
| ln(Fecundity) | 0.16* | −0.19 | – | | | | |
| ln(Maximum length) | 0.65 | −0.14 | 0.11 | – | | | |
| ln(Depth) | 0.22 | 0.08 | 0.01* | 0.30 | – | | |
| Trophic level | 0.27 | 0.03 | −0.06 | 0.50 | 0.38 | – | |
| Optimal temperature | −0.08 | −0.01 | 0.23 | 0.06 | −0.10* | −0.04 | – |

and pelagic species; however, if seabed type is also taken into account apart from open sea species hard substrate ones had higher values. Maximum length had significant variation across six categorical traits, with the most striking being the highest values in pelagics, pelagic spawners and non-migratory species. Regarding depth, eurybathic species and benthic spawners were found to occur deeper, herbivore diet/grazing behavior and euryhaline species were found shallower. The highest trophic level was naturally found in piscivorous species, ambushing predators (and mobility type) and eurybathic species. Optimal temperature was found to be higher in species of high and medium mobility, stenothermal species and species of tropical distribution.

The trait co-occurrence analysis documented 170 (17.4%) positive, 183 (18.8%) negative, and 622 (63.8%) random modality associations (Fig. 4). The modalities with the highest number of positive co-occurrences are associated with the pelagic (e.g., free exposure with 13 positive co-occurrences) or the benthic way of living (e.g., benthic spawning habitat with 14 positive co-occurrences, flat body shape and benthic habitat type with 13). The modalities with the highest number of negative co-occurrences are associated mostly with the pelagic way of living (atractoid body shape, sociability schools, seabed type water column and pelagic habitat all had 15–17 negative co-occurrences). Additionally, deep body shape and solitary sociability had relatively many positive and negative co-occurrences. Relatively rare trait categories (e.g., tropical distribution, autumn spawning) had a small number of co-occurrences. At the scale of entire traits, body shape, depth range, mobility and exposure had the highest cumulative positive co-occurrences of all their modalities, while body shape, mobility, habitat type and exposure had the highest cumulative negative cooccurrences. The lowest number of cumulative co-occurrences (both negative and positive) was found for the traits: hermaphroditism, diet, seasonal migrations, seabed type and optimal temperature.

The Hill-Smith ordination resulted into many axes, each explaining a relatively small variance percentage (in Fig. 5, the first two axes presented explain 19% of cumulative variance). The dendrogram resulting from the hierarchical clustering of the species coordinates (Fig. 6), if cut at a high value of dissimilarity (e.g., 70%, not shown in the figure) distinguishes three major groups of pelagic, benthopelagic and benthic species and

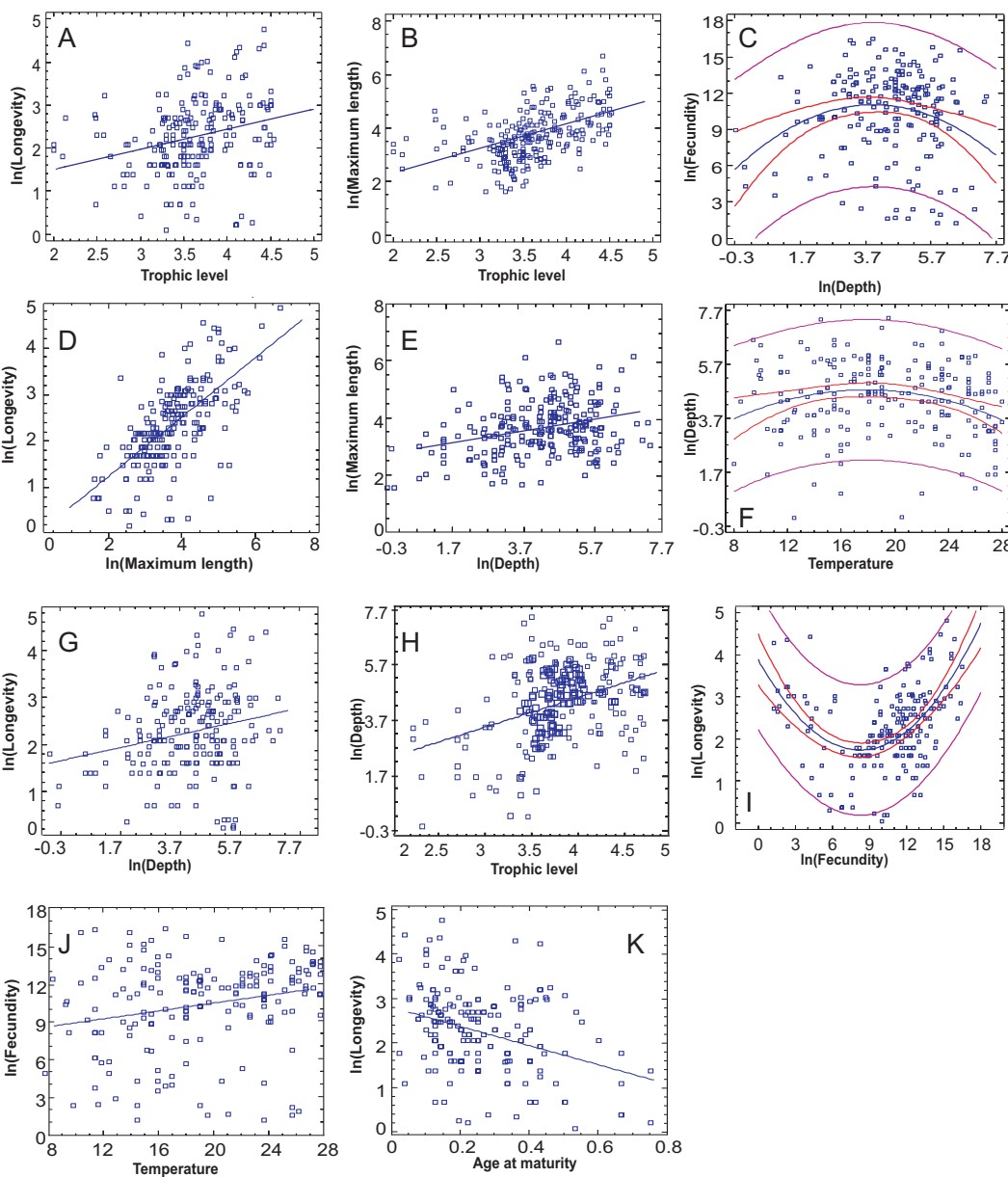

**Figure 3** **The statistically significant correlations or non-linear relationships between continuous traits after incorporating the Bonferroni correction.** (A) Trophic level-Longevity, (B) Trophic level-Maximum length, (C) Depth-Fecundity, (D) Maximum length-Longevity, (E) Depth-Maximum length, (F) Temperature-Depth, (G) Depth-Longevity, (H) Trophic level-Depth, (I) Fecundity-Longevity, (J) Temperature-Fecundity, (K) Age at maturity-Longevity.

the associated traits. At lower dissimilarities (42%), six main groups are extracted. As these groups include in some cases a wide range of species with very different functional roles (e.g., group A includes pelagic species spanning from swordfish and tuna to anchovy and sardine) it was deemed necessary to determine grouping in lower dissimilarity levels (13.5% as shown in the figure) resulting in the definition of 18, more homogeneous internally,

**Table 4  Results of the statistical analyses between continuous and categorical traits.** Test (A: ANOVA, T: $t$-test, K: Kruskal–Wallis, M: Mann–Whitney) indicated as superscript next to result.

| Trait | ln (Longevity) | Age-at-maturity | ln (Fecundity) | ln (Maximum length) | ln (Depth) | Trophic level | Optimal temperature |
|---|---|---|---|---|---|---|---|
| Gonochorism | −1.21$^T$ | −339.50$^M$ | 391.5$^M$ | −75.50$^M$ | 214.50$^M$ | 0.21$^T$ | −1.23$^T$ |
| Body shape | **17.71$^K$ | 1.81$^A$ | ***24.28$^K$ | ***8.58$^A$ | 8.67$^K$ | 7.05$^K$ | **4.04$^A$ |
| Habitat type | 0.79$^A$ | 0.8$^A$ | ***13.43$^K$ | ***9.38$^A$ | 0.13$^A$ | **10.32$^A$ | *3.34$^A$ |
| Seasonally migratory | 629.50$^M$ | −0.34$^T$ | *−2.45$^T$ | ***−5.27$^T$ | 0.10$^T$ | −1.61$^T$ | −0.73$^T$ |
| Distribution | 2.08$^A$ | *2.75$^A$ | 1.52$^A$ | 2.00$^A$ | *3.42$^A$ | 0.83$^A$ | ***12.17$^A$ |
| Seabed morphology | 0.96$^A$ | 0.48$^A$ | ***16.97$^K$ | 2.57$^A$ | 1.58$^A$ | 3.57$^A$ | *4.22$^A$ |
| Diet | ***6.16$^A$ | 0.57$^A$ | 10.58$^K$ | ***30.41$^A$ | ***49.43$^K$ | ***143.7$^K$ | 1.65$^A$ |
| Feeding behaviour | ***27.29$^K$ | 0.72$^A$ | 2.65$^A$ | ***42.98$^K$ | ***21.38$^K$ | ***76.18$^A$ | 1.64$^A$ |
| Spawning period | 4.65$^K$ | 0.83$^A$ | 0.65$^A$ | 1.13$^A$ | *3.00$^A$ | 0.86$^A$ | *2.49$^A$ |
| Spawning habitat | **1,276.50$^M$ | 1.03$^T$ | **−2.85$^T$ | ***1,664$^M$ | ***1,762$^M$ | −1.53$^T$ | −0.8$^T$ |
| Depth range | 0.11$^T$ | **−1,222$^M$ | −1.18$^T$ | *2.5$^T$ | *** −63.65$^M$ | ***3.53$^T$ | −0.76$^T$ |
| Temperature range | 1.59$^T$ | −0.49$^T$ | −1.83$^T$ | −0.95$^T$ | 0.03$^T$ | 681.5$^M$ | ***−4.8$^T$ |
| Salinity range | 0.81$^T$ | −1.3$^T$ | 1.37$^T$ | 1.51$^T$ | ***−5.47$^T$ | −1.58$^T$ | −0.31$^T$ |
| Sociability | 0.23$^A$ | 1.98$^A$ | **12.72$^K$ | 0.18$^A$ | 2.24$^A$ | 7.6$^K$ | 2.16$^A$ |
| Exposure | 0.11$^A$ | *3.78$^A$ | 2.22$^A$ | 0.81$^A$ | *4.25$^A$ | 0.89$^A$ | *3.02$^A$ |
| Mobility | **4.31$^A$ | 0.82$^A$ | **4.23$^A$ | **4.44$^A$ | *3.48$^A$ | **15.25$^K$ | **4.79$^A$ |

**Notes.**

*, $p < 0,05$.; **, $p < 0,01$.; ***, $p < 0,001$..

Statistically significant differences between groups (categorical traits) without taking into account the Bonferroni correction are denoted in red color, while these incorporating the Bonferroni correction are indicated in green.

functional groups. The traits characterizing the six major and the 18 minor functional groups are presented in Table S2. From both Figs. 5 and 6 and the supplementary table, it is evident that while the coarse distinction (six groups: A–F) can indicate the major groupings of nektonic organisms, the grouping at a higher level of similarity can highlight major functional components of the nekton within the ecosystem, like small pelagic species (Group 1) or herbivorous fish (Group 3).

## DISCUSSION

The nekton functional groups identified here can be a useful tool to study the ecology of the Mediterranean Sea, both in analyses using empirical data and in simulation models that utilize functional groups to operate (e.g., Ecopath -*Pauly, Christensen & Walters, 2000*). Despite the fact that, from the clustering of Fig. 5, the initial choice would be to divide the dendrogram either in the three major clades (pelagic, benthopelagic and benthic groups) or in the six groups (A-F) identified at 42% dissimilarity level, it is more informative and reasonable to use a lower dissimilarity that leads to the distinction of groups with different actual functional roles (e.g., 1: small pelagics, 3: vegetation grazers) or with higher homogeneity in traits (like the division the pelagic group A that on average had large size into the species with small size in group 1 and those with larger size in group 2). As nekton can be generally expected to occupy a variety of ecological niches because

**Table 5 Summary of the main findings of the significant relationships between continuous and categorical traits.** For pairwise contrasts between trait categories see Fig. S1.

| Continuous trait | Categorical trait | Main findings |
| --- | --- | --- |
| Longevity | Body shape | Highest in flat species, lowest in rounded species |
| | Feeding type | Highest in ambushing & active predators, lowest in grazers |
| | Spawning habitat | Higher in pelagic spawners |
| | Diet | Highest in piscivorous species, lowest in zooplanktivorous-zoobentivorous |
| Fecundity | Body shape | Higher in atractoid and deep-bodied, lower in flat and long species |
| | Habitat type | Highest in pelagic species, intermediate in benthopelagic, lowest in benthic |
| | Seabed type | Highest in the open sea and over hard substrate, lowest over soft substrate |
| Maximum length | Body shape | Higher in long, atractoid and flat body shape, lowest in rounded body shape |
| | Habitat type | Highest in pelagic species, intermediate in benthic, lowest in benthopelagic |
| | Seasonal migrations | Higher in non-migratory species |
| | Diet | Highest in piscivorous species |
| | Feeding type | Highest in active & ambushing predators, lowest in grazers |
| | Spawning habitat | Higher in pelagic spawners |
| Depth | Depth range | Eurybathic species deeper |
| | Diet | Herbivores most shallow, piscivores and zoobenthivores deeper |
| | Feeding type | Active predators deepest, grazers shallowest |
| | Spawning habitat | Benthic spawners deeper |
| | Salinity range | Euryhaline species in shallower depth |
| Trophic level | Diet | Herbivores have lowest trophic level, piscivores the highest |
| | Feeding type | Highest in ambushing predators, lowest in grazers |
| | Habitat type | Highest in pelagic species |
| | Depth range | Eurybathic species have higher trophic level |
| | Mobility | Ambushers have highest trophic level |
| Optimal temperature | Mobility | Higher in species of high & medium mobility, lower in small mobility species |
| | Temperature range | Higher in stenothermal species |
| | Distribution | Highest in tropical species, lowest in temperate |

of the variety in species traits like size, habitat use and diet, it is not surprising to have more functional groups than those found in zooplankton (e.g., *Benedetti et al., 2018*). Further steps examining only traits related to resource use could shed light into potential inter-specific competition relationships and niche overlap (M. Koutsidi unpublished data).

*Violle et al. (2017)* underline the importance of functional rarity and the ecology of outliers as complementary to the concept of the traditional taxonomic rarity. While rare species may share traits with more abundant ones, in the case that the traits (and resulting functions) are rare, loss of the species that possess them may significantly alter ecosystem functioning (*Jain et al., 2014*). This is also relevant to the concept of keystoneness (a
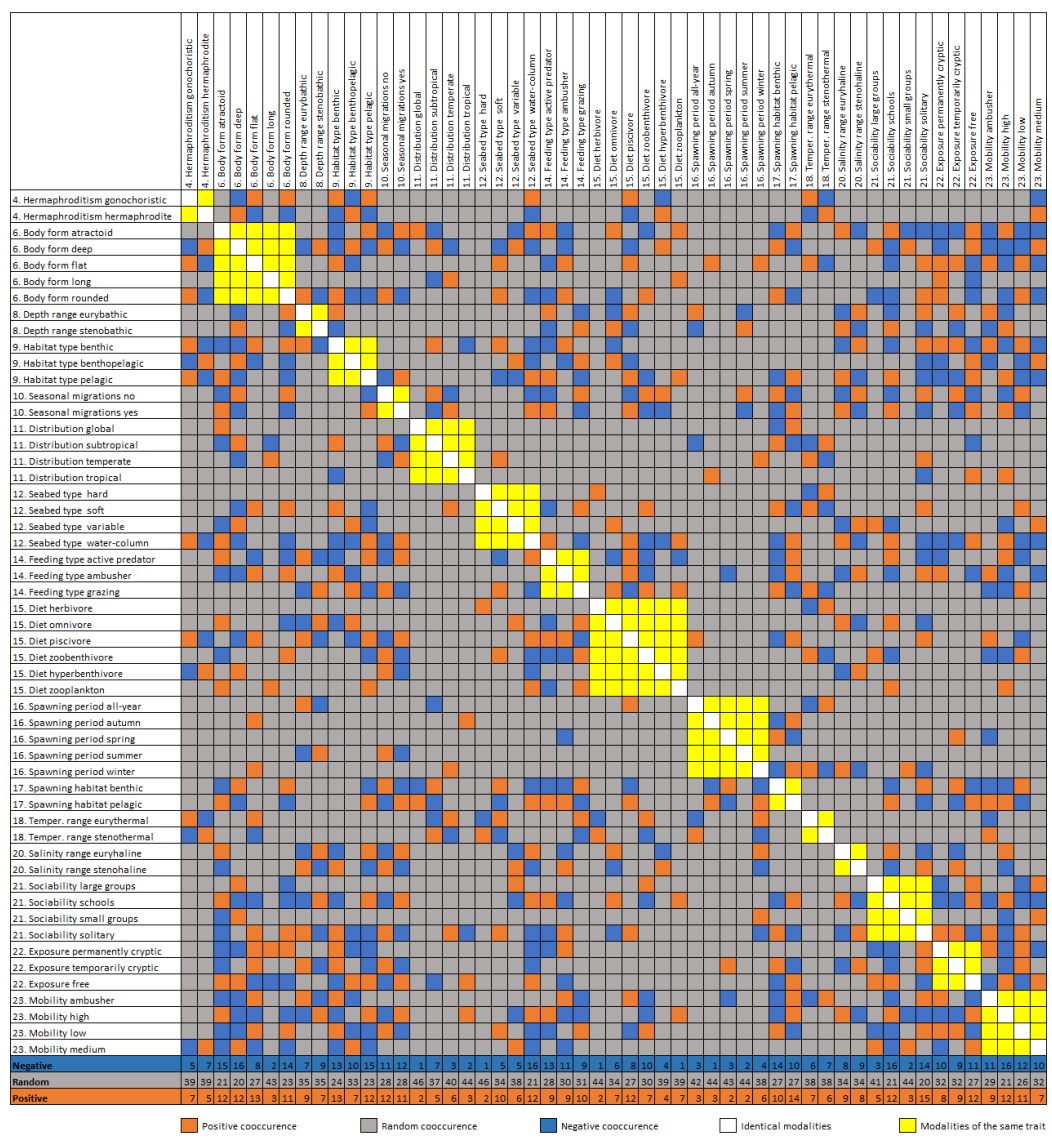

**Figure 4  Diagonal matrix of positive, random, and negative co-occurrence between the trait category/modality pairs.** Cumulative number of positive, negative and random co-occurrences are also provided for each trait category.

keystone species being a species whose importance for its community is disproportionally high in comparison to its abundance *Bond (2001)*. Here we document the rarity (even below 5% of the species total) of autumn spawning (but also winter and all year spawning) as well as that of herbivory. Herbivory is anyway considered a crucial aspect of ecosystem functioning as alterations in herbivory may cause community phase shifts where the main habitat-formers are lost or substituted by very different ones: *Vergés et al. (2014)* have documented this regarding the populations of *Siganus* sp. that have invaded the Mediterranean Sea, but there are also examples from coral reefs (*Hughes et al., 2010*) and temperate algal forests (*Steneck et al., 2002*). Thus, it is important to maintain and

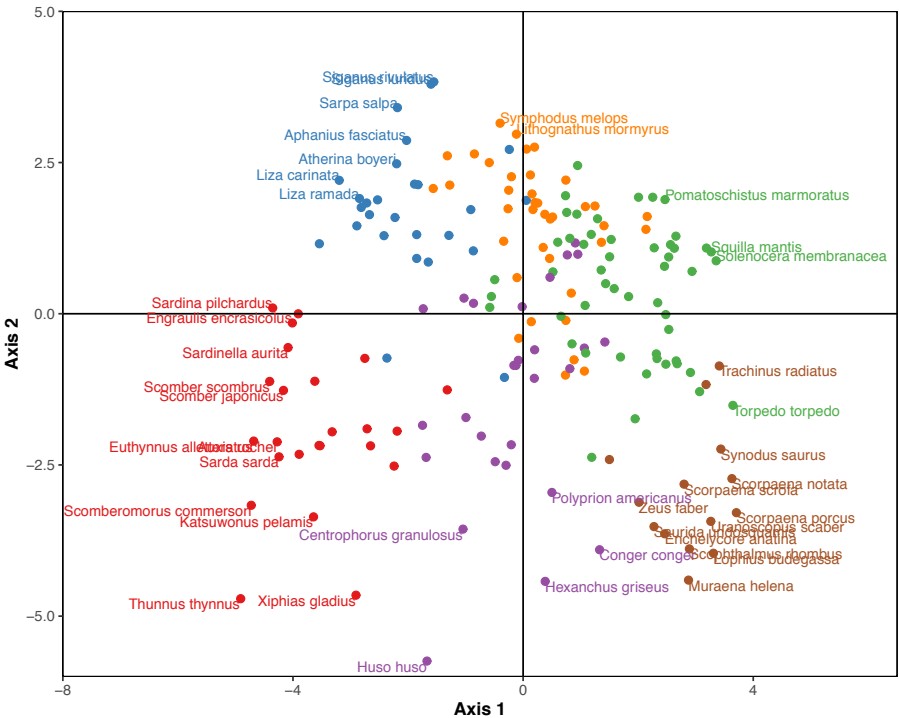

**Figure 5** **Species positions across the first two major axes (explaining 19% of cumulative variance) of the Hill-Smith ordination.** Species major groups (at 42% dissimilarity) are indicated by colours corresponding to the dendrogram of Fig. 6 and some species names are also provided.

regulate the abundance and the rate of renewal of this trait, especially taking into account the fact that some of the invading species (e.g., *Siganus* sp.) in the Mediterranean are herbivorous and competing with resident species (M. Koutsidi, unpublished data). Similar rarity of herbivory has been documented by *Beukhof et al. (2019)* for marine fish from North Atlantic and Northeast Pacific continental shelf seas (in that work the importance of piscivory is much lower, however the dominant modality is that of generalist feeding, which was not used in the dataset of the present work). It is true that herbivory is also carried out by other, benthic, biota (e.g., echinoderms); however, they have different traits (e.g., mobility) that may change this function. Successful seasonal spawning, like autumn and winter spawning -and also the success of the recruitment that follows it- may be prone to various environmental factors, also possibly affected by anthropogenic effects like fisheries (that are characteristically seasonal in the Mediterranean) or climate change that may decrease the duration of the window-period suitable for spawning (Table 1). The above indicate the clear need for a holistic assessment evaluation of traits including all biotic elements of the ecosystem (plankton, nekton and benthos).

The rarity of other traits like hard seabed type preference may be related to the relatively small extent of this habitat type in the marine environment. Similarly, the low occurrence of long body shape may be related to the scarcity in characteristics of the habitat (e.g., structurally complex habitat for long body shape which here was indeed found to cooccur

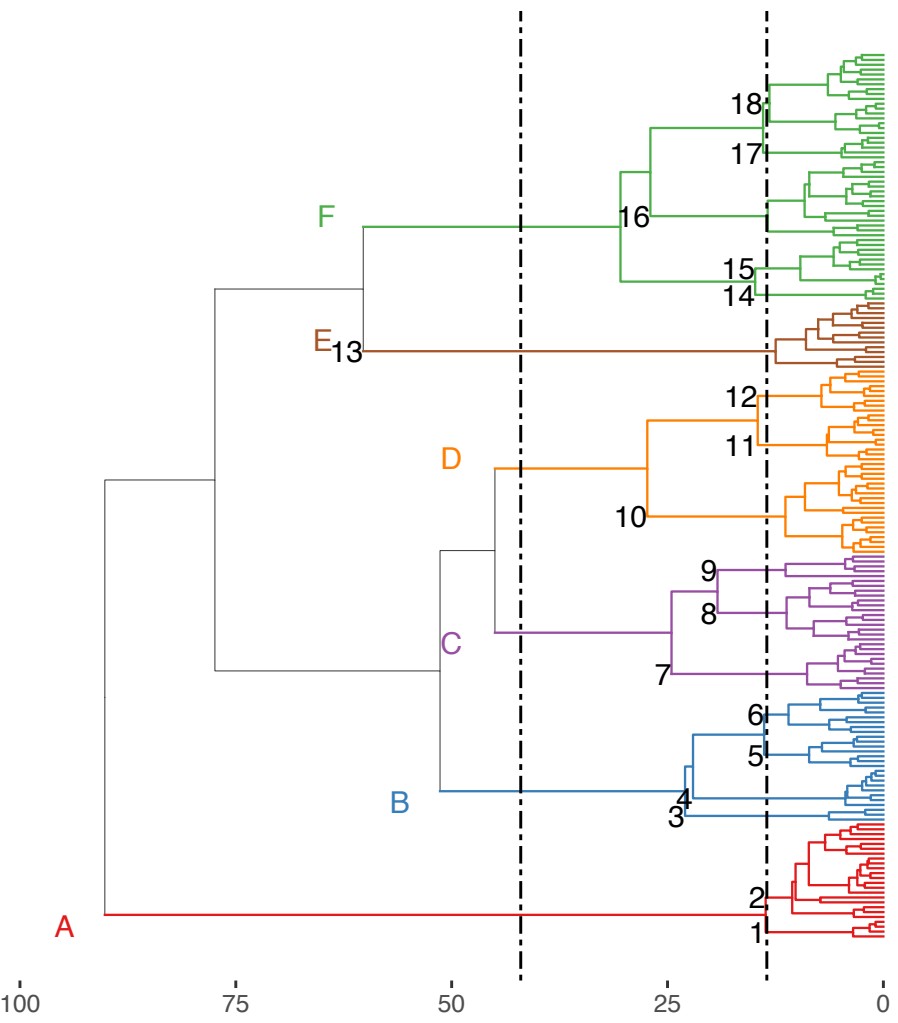

**Figure 6** Hierarchical clustering of nektonic species using the coordinates of the 11 major axes of the Hill-Smith ordination and functional group identification at two dissimilarity levels: 42% (Groups A–F) and 13% (Groups 1–18).

with cryptic exposure). However, the similarly rare flat body shape cooccurs with soft seabed preference which is a relatively common trait, indicating that trait relationships may be less straightforward. Even though the body-shape trait categories in *Beukhof et al. (2019)* for North Atlantic and Northeast Pacific are not exactly similar, the long and eel-like body shapes are overall more common there, while the deep body shape tends to be more common in the Mediterranean.

It is important to note that trait rarity should not only be evaluated at the species level, but also weighted with species abundance or biomass to indicate the actual "abundance" of traits in the ecosystem (*Violle et al., 2017*). E.g., the planktivorous diet trait category may be rare if evaluated using the number of species but very abundant as the small pelagic or benthopelagic species that possess it have very high abundances. Still, the fact that it is shared by only a handful of species may be a risk for ecosystem functioning, especially
taking into account the fact that these species are known to have interannual abundance fluctuations. Even more so, some of these species like the European anchovy *Engraulis encrasicholus* and the European pilchard *Sardina pilchardus* are under intense fishing pressure and have been shown to be affected by climate (*Vasilakopoulos et al., 2017*).

The present work confirms that the combinations of functional traits in species of Mediterranean fisheries resources are not random (e.g., *Jennings et al., 2002*), since the evolutionary process has provided species with certain trait combinations (*Gislason et al., 2010*). Here we reconfirm that species with a long lifespan also have large body size (*Vila-Gispert, Alcaraz & García-Berthou, 2005*). *Charnov, Gislason & Pope (2013)* note that species only grow to a large body size if natural mortality rates are low, thus their life span is long. The examination of the distribution of species in trait space using a randomization approach similar to *Díaz et al. (2016)* could provide more definitive results on the non-random distribution of traits.

In the present work, longevity and age at maturity are negatively correlated, in ostensible disagreement with previous studies documenting a positive relationship (*Froese & Binohlan, 2000*; *Jarić & Gavčić, 2012*). Contrary to these works (where age at maturity is expressed in years), here age at maturity was examined as a percentage of the species lifespan. Indeed, if age at maturity is expressed in years in our dataset, there is a positive correlation with longevity ($r = 0.72$, $p < 0.001$); however here we had intended to determine how early or late a species matures regarding its life duration. Therefore, species with a short life span tend to mature relatively late in their lifetime. This can be interpreted, if we take into account that even a short-lived species needs to have completed an amount of growth to reach a minimum size and biomass for reproduction (*Beverton, 1963*).

The positive relationship between size and trophic level found in this study has also been documented in other works (e.g., *Romanuk, Hayward & Hutchings, 2011*). Trophic level increases with increasing body size, because most predators are larger than their prey (*Kaiser et al., 2005*). *Jennings et al. (2002)* also point out that, in benthic communities, trophic level and body mass of species have a strong relationship. Additionally, in this study we found a relationship between maximum body length and average depth species distribution. Finally, in this study, non-linear (polynomial) relationships between traits are indicated perhaps because of the existence of sharks and rays in the dataset. Especially their high longevity and low fecundity, result in deviations from the linear pattern.

The co-occurrence analysis indicates some positive and negative associations between pairs of trait modalities. The main characteristics of small pelagic fish (e.g., *Sardina pilchardus, Engraulis encrasicolus*) have the highest number of positive and negative associations with the other modalities. The traits of the family Sparidae, such as deep body shape, hermaphroditism (e.g., *Sparus aurata*) and grazing feeding (e.g *Sarpa salpa*), were found to have a relatively high number of positive associations with other modalities. The detection of relationships between traits is important not only as a way to explore the relationships of characteristics shaping life, but also because it could be useful to predict the possible effects of anthropogenic pressures on these traits. For example, climate change can be expected to favour thermophilic species, thus traits related to high optimum temperatures (high fecundity, deep body shape, high mobility) may be favoured as well.

*Koutsidi et al. (2016)* have documented the removal of specific traits by fisheries; this could also result in modification of traits composition depending on fisheries management. Furthermore, *Edeline et al. (2007)* have demonstrated that anthropogenic stressors like fishing may act as selective pressures favouring specific traits (in their study slow growth) that are heritable. *Law (2000)* mentions some examples of phenotypic changes induced by fisheries and discusses the heritability of these traits. *Mousseau & Roff, (1987)* have documented higher heritability of morphological traits to life-history traits. *Law (2000)* indicates that heritability in the range of that demonstrated for life history traits, despite being relatively small is still enough to cause substantial selection responses within a small number of generations. The existence of fisheries-induced evolution is now well-documented in traits like growth and maturation (*Kuparinen & Merilä, 2007*; *Enberg et al., 2012*) and complex interactions between natural and anthropogenic selection factors acting in opposite directions may exist (*Edeline et al., 2007*).

The current work has some findings differing from those of *Koutsidi et al. (2016)*, as e.g., the associations of depth with fecundity and optimal temperatures documented here are not reported there. This can be a result of the inclusion of many more species here, but also because of the treatment of traits according to the variable type here and not as categories in all cases. Many rare traits identified like ambushing feeding behavior, flat and long body shape, autumn spawning, cosmopolitan and tropical distribution and low trophic level and fecundity are common in both works while others like long lifespan and distribution in deep water are novel here and again indicate that continuous traits are better analyzed as such.

Naturally (and as shown also here) traits are related. This is not only with regard to life strategies shaped by evolutionary processes (e.g., larger species having longer life duration too), but also as they may be relevant (e.g., diet and trophic level). Still, apart from the trait affinity, and despite the fact that there is the tendency to try to include only functional traits in analyses (i.e., more relevant to life cycle and resource use), different traits may still convey different information and still vary (e.g., as shown here piscivorous fish tend to have higher trophic level, but may still span a range of trophic level values depending on their prey). Furthermore, even relevant traits may incorporate information with different significance for ecosystem functioning or resilience (see e.g., the significance of traits regarding tolerance range for variables like temperature and depth in comparison to the optimal values of these factors in Table 5). Thus, depending on the research question, some, even relevant traits can be useful for the evaluation of findings.

Here, we have not limited our analyses to the 13 functional traits (*Violle et al., 2007*) of our dataset, but also include ten traits that would be rather characterized as ecological (*Beauchard et al., 2017*). The determination of relationships between functional traits can indicate affinities derived from phylogenetic constraints or indicate life strategies as they have been shaped by evolution. Relationships between ecological traits can indicate the major aspects of organismal distribution in time and space, while relationships among functional traits and ecological traits can indicate the potential environmental filtering that acts on functional traits. It should be noted that, apart from traits that are not related with many others (like age at maturity and fecundity), most traits were found to have

significant relationships with both functional and ecological other traits independent of themselves being of a functional or ecological significance, thus showing some level of environmental filtering on functional aspects; however these relationships should also be examined weighted by abundance or biomass. Anthropogenic stressors may also affect differently these two trait types, as e.g., fisheries act on both functional (e.g., life-cycle, behaviour) and ecological (e.g., habitat) aspects of a population, while the effect of climate is primarily ecological (e.g., temperature optimum and range), but both stressors can have indirect effects on other, related, traits.

In this work, we have assigned each species to a single modality per trait. However, it is true that traits can vary across individuals or populations (*Violle et al., 2017*) and in some cases a species could be assigned to have more than one modality in a quantitative way (e.g., a species spawning from December to April is here assigned to spawn in winter, while alternatively it could be assigned as spawning in 100% of winter months and 33% of spring) using fuzzy coding. Such an approach should be evaluated in future works not only regarding the correct assignment of information to modalities, but also to account for species plasticity thus rendering the analyses more realistic (*Chevenet, Dolédec & Chessel, 1994*). In the same context, while we used one value as representative of a species, trait values may vary across the entire species distribution or the region examined (e.g., there may be differences between the western and eastern Mediterranean, especially in some continuous traits). Additionally, while in many cases, trait datasets are assembled only regarding the mature stages of a species, it is true that juveniles can possess different trait values (e.g., diet). Furthermore, the major part of abundance/biomass of a population may belong to the juvenile stages. An interesting approach would be to enrich trait datasets with distinct trait values between juveniles and adults. Data from monitoring programs regarding population structure (e.g., through the length distribution that is typical in fisheries monitoring) could be used along with community composition for a more realistic depiction of actual trait space occupied. This would be a very valuable expansion of trait datasets and their usage; however, information on juvenile stages might be hard to obtain, especially for non-commercial species, whose biology is in some cases not fully documented. Finally, basic biological research (especially for the largely unknown deep-sea fish) and assembly of even more extensive trait datasets in terms of species, at least with a focus to the traits that are more representative of functional diversity would help fill the gaps of the trait database, especially with the aim to focus on understudied Mediterranean ecosystems.

The question as to which traits constitute more fundamental information for ecosystem functioning still pertains. As ecosystem functioning is related to the transfer of energy and material and the regulation and maintenance of ecological processes (*Naeem et al., 1999*; *Bremner, Rogers & Frid, 2006*; *Paterson, Defew & Jabour, 2012*), traits related to trophic interactions like diet and trophic level (i.e., effect traits according to *Lavorel & Garnier, 2002*; *Violle et al., 2007*; *Suding et al., 2008*) are of direct significance. Traits affecting these interactions in space and time (mainly response traits according to the same authors), such as habitat and depth distribution, migration, spawning season and even fecundity are also relevant (Table 1). In plants, it has been shown that, in cases where response traits are also

effect traits, there can be loss of ecosystem function (*Suding et al., 2008*). Moving from the scale of nekton to the scale of the entire ecosystem, i.e., incorporating biotic components like plankton and benthos, as functional traits like maximum length and diet/trophic level are key for energy transfer they should be prioritized to be taken into consideration. *Pecuchet et al. (2019)*, investigating ecosystem-wide functional reorganization in the Baltic Sea by examining multi-trophic communities indicate that diet or type of feeding traits are in many cases relevant for many of the groups examined; however, they underline that different traits are involved and demonstrate diverse dynamics among areas. *Litchman, Ohman & Kiørboe (2013)* refer to the significance of body size as a trait that is related to many others that are worth monitoring for ecosystem studies (like growth rate, stoichiometric requirements, grazing rate and trophic niche breadth), but underline that the choice on which traits to monitor ultimately relies on the questions asked. In this regard, ecological traits regarding spatial and temporal distribution (habitat and seabed type, environmental variable ranges and optima) or occurrence of critical life cycle events like spawning (spawning period and habitat) could also be useful in supplementing this information to describe the spatio-temporal cooccurrence of the different elements for a holistic evaluation of the marine ecosystem.

With the global ocean being under a multitude of anthropogenic effects (e.g., *Crain et al., 2009*), it is crucial to identify traits that are significant for monitoring human induced alterations in the structure and dynamics of the marine ecosystem. These traits are not only important as descriptors of the marine community (see e.g., the "mean temperature of the catch"–*Cheung, Watson & Pauly, 2013*) useful in monitoring, but should also be maintained to some minimum levels, to avoid function loss or the creation of too many empty ecological niches that could more easily be colonized, e.g., by alien species (*Givan et al., 2017*). Thus, regarding climate change, traits like optimal temperature and temperature range are significant, as communities with a diversity of thermal affinities and narrow ranges of thermal tolerance are more sensitive to climate change (*Burrows et al., in press*); yet all range-type traits and also distribution seem to have important implications for ecosystem dynamics and resilience. Regarding fisheries effects, size (also because of the various significant relationships it has with other traits and its implications for management) is a crucial trait. Other core biological traits like longevity, fecundity and age at maturity are also important for fisheries management, but also some behavioural traits relevant for the interaction of nekton with fishing gear may bear some importance. *Lavorel & Garnier (2002)* and *Violle et al. (2007)* suggest that traits whose attributes vary as a response to changes in environmental conditions are response traits, while traits determining the effect of an organism to the environmental conditions, biotic or abiotic are effect traits. While this distinction is both valid and useful, *Lavorel & Garnier (2002)* and *Suding et al. (2008)* have already indicated that a trait can act as both response and effect; in Table 1 we also document that, in some cases, a response trait (e.g., optimal temperature), having been affected by change in environmental conditions may in turn act as an effect trait altering subsequently the dynamics or composition of the community. This is especially important regarding the direct and indirect effects of anthropogenic stressors like fisheries and climate change. The relationships between traits documented here and elsewhere may

help explaining the dynamics of nektonic communities (*Marquez et al., 2019*) and marine ecosystems and also make predictions for future scenarios, as anthropogenic stressors may alter trait composition indirectly (*Henderson et al., 2019*), through selection of associated traits.

## CONCLUSIONS

Relationships between biological traits have long been studied to investigate how evolution has shaped life into form and function and how characteristics are combined into life strategies. However, the study of nektonic trait distribution and combinations presented here can be useful to elucidate trait interactions significant for indirect alterations of ecosystem functioning, especially today, when the marine environment is under a multitude of anthropogenic stressors that can act on specific traits. The documentation of rare traits (like winter and autumn spawning, herbivory, very low or high size or fecundity, hard substrate type and ambushing predation) in species together with the appraisal of the significance of traits at various scales indicates aspects of crucial importance that need to be preserved. Under a more synthetic scope, nektonic functional groups are broadly determined around major aspects of habitat use (pelagic, benthopelagic or benthic), but can be distinguished in more detail showing affinities among and between functional and ecological traits that can be used in the future to understand nektonic communities and model ecosystems. In any case, the documentation of a multitude of relationships between functional and ecological traits found here indicates how the environment, through the delimitation of species distribution in space and time depending on their traits, can filter functional traits, while the validation of functional trait associations hints at functional interdependencies determined by evolution. The findings documented here highlight the traits that should be evaluated and monitored in the future both at the level of nekton or in combination with other major ecosystem components for the assessment of ecosystem functioning and those that should be maintained to ensure ecosystem resilience.

## ACKNOWLEDGEMENTS

The authors would like to thank the editor as well as the three reviewers for their valuable suggestions and comments that helped to significantly improve the manuscript. We would also like to thank the participants of the International Council for the Exploration of the Sea Working Group on Comparative Analyses between European Atlantic and Mediterranean Marine Ecosystems to Move Towards an Ecosystem-based Approach to Fisheries (ICES WG COMEDA) 2018 meeting for comments on preliminary results.

### Funding

This research was supported by Grant 56690000 from the Research Committee of the University of Patras via ''K. Karatheodori'' program. The funders had no role in study design, data collection and analysis, decision to publish, or preparation of the manuscript.

## Grant Disclosures

The following grant information was disclosed by the authors:
Research Committee of the University of Patras via ''K. Karatheodori'' program.: 56690000.

## Competing Interests

The authors declare there are no competing interests.

## Author Contributions

- Evangelos Tzanatos analyzed the data, prepared figures and/or tables, authored or reviewed drafts of the paper, conceived the work and acquired funding, and approved the final draft.
- Catherine Moukas and Martha Koutsidi analyzed the data, prepared figures and/or tables, authored or reviewed drafts of the paper, assembled data on biological traits, and approved the final draft.

## Data Availability

Data is available on Figshare: Koutsidi, Martha; Moukas, Catherine; Tzanatos, Evangelos (2019): Koutsidi, Moukas, Tzanatos: 23 biological traits of 235 species. figshare. Dataset. https://doi.org/10.6084/m9.figshare.11347406.v1.

## Supplemental Information

Supplemental information for this article can be found online at http://dx.doi.org/10.7717/peerj.8494#supplemental-information.

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
