# Peer review of "Mediterranean nekton traits: distribution, relationships and significance for marine ecology monitoring and management"

_PeerJ, doi:10.7717/peerj.8494_

## Round 0.1 · original submission · Major Revisions

Two of the three reviews suggested major revisions. While the proposed manuscript will certainly be useful to the community of ecologists working on functional diversity of fish, it should be based on more established definitions. The discussion should also be further developed.

·

Basic reporting

The paper includes a sufficient introduction as well as a broad discussion and provides the relevant literature references.
Article structure, tables, figures and raw data conform to scientific claims. The figures are relevant and professional. The submitted paper is self-contained and sound. The results are relevant, but the aims should be complemented and oriented to the title and the discussion respectively (more details see below).
The text is written in clear and unambiguous English, but the word usage should be improved once more by a native speaker.

Experimental design

No comment / not applicable.

Validity of the findings

The study is meaningful, never before there was assessed such a broad selection of traits of Mediterranean fish or other taxa of nekton. I agree with the authors that an investigation of patterns and relationships between these traits is important, especially for the Mediterranean, because the latter is a region that hosts a high biodiversity and has a long history of human influence.
The underlying data have been provided. The authors use simple statistics but it is applicable to the aims of this paper, robust and sound. The conclusions are well stated.

Additional comments

Dear authors,
This paper is well written in large parts, brings novel results and conclusions, and therefore it is eligible for publication in my opinion. However, there are several issues that should be improved.
The most important are following:
(i) Several references cited in the text are lacking in the reference list and vice versa. Please check all references and citations carefully.
(ii) The link to the row data of species and traits given in line 103 does NOT lead to your row data.
(iii) The conclusions (lines 332 – 44) are rather a summary of the discussion than 'conclusions'. This paragraph should be rewritten or omitted.
(iv) Any information about the examined animals (e.g., how many fishes, cephalopods, decapods a.s.o.), would be welcome in the 'Material and Methods'.

(v) As already mentioned above, you stated that the aims of this paper are: (a) to determine whether there are rare or important traits by evaluating the distribution of traits and (b) to detect whether there are undocumented relationships between pairs of biological traits.
But the paper includes much more: a number of findings have significant implications concerning the potential effects of climate change (e.g. through the relationships of the trait of optimal temperature) or habitat loss (derived from the relationships of traits related to tolerance ranges) or the impact of fisheries.
Thus, you should add to your aims that selected 'rare traits' are discussed in relation to climate change and to species' reliance to human impact which also would correspond to the title of the paper. Because you examined many more traits (and even more species) than comparable earlier studies, this part of the discussion is important in my opinion, making the paper especially interesting.

(vi) Does Gambusia affinis play any role in the Mediterranean Sea, or why was it included in your analysis?

You will find some additional remarks in the pdf file.

I hope that these recommendations will be comprehensible and contribute to improve your article.

With kind regards,
Erik Arndt

·

Basic reporting

In general, the manuscript is well written; the general context is well established and is based on a good knowledge of literature. I greatly appreciate the table 1 which establish the links between traits and the responses of organisms, population, community or ecosystem levels. However, a precise definition of what a trait is would be very useful to avoid confusion and strengthen the scope of the manuscript. Currently, the term “trait” is used in multiple context and the use of different adjectives can generate a misunderstanding. See for instance the use of “biological trait”, “ecological trait”, “functional traits”, “response trait”, “effect trait”…. In this context, Violle et al. (2007) proposed a common framework that can be used such as:
• Trait: Any morphological, physiological or phenological feature measurable at the individual level, from the cell to the whole-organism level, without reference to the environment or any other level of organization.
• Functional trait: Any trait which impacts fitness indirectly via its effects on growth, reproduction and survival.
In most literature, functional traits are then divided into 4 types: life history traits, morphological traits, physiological traits and behavioural traits. Ecological traits that refer to environmental preferences are generally excluded even if Costello et al. (2015) in PeerJ 3, e1201 proposed 10 ”priority” biological traits that include both functional and ecological traits.
Personally, I agree with Violle’s definitions which exclude ecological traits. In your paper, many traits are in fact ecological traits which can lead to confusion in interpretations. I therefore suggest that we clearly separate functional traits from ecological traits. Thus, the study of relationships between functional traits alone will make it possible to identify relationships that result from phylogenetic constraints or redundant information between traits (e.g. diet and trophic level), while relationships between functional traits and habitats will identify the influence of the environmental filtering on functional traits. Such a dichotomy should make it easier to discuss the results obtained and to better understand the effects of climate change or fishing on functional traits.
In any case, it is essential to clearly define the terms used in the manuscript.

The figures are relevant but would benefit from more detailed legends; some information is also missing. In Figure 1B, there is no title for the x-axis that corresponds to the rank of species according to the values of the line ranges. In Figure 2, no titles are given to the abscissa and ordinate axes. In parallel with the list of traits in table 2, a definition of each trait should be useful, may be only as supplementary material. In table 3, as the Pearson correlation coefficients matrix is a triangular matrix, only half of the matrix could be provided. It is also misleading to provide non-significant value when the linear regression is not well-adapted. I suggest to replace in the matrix the Pearson correlation coefficients by the regression coefficient.

Raw data (i.e. the database containing the 23 quantitative and categorical traits of fish) is provided as an Excel file that can be downloaded. This database represents a remarkable amount of data compilation work (979 references have been used) that will be extremely useful in the future for all researchers working on biological traits of fish in the Mediterranean Sea. However, I regret that the geographical origin of the data is not explicitly specified as some traits may vary from region to region, or from one habitat to another one. This issue could be also discussed.

Experimental design

The manuscript is a research article which fits well the scope of PeerJ. It is a continuation of an earlier article published by Koutsidi et al (2016) which addressed the same questions on a smaller dataset that only considered traits as categorical variables. The article provides then an analysis of biological traits in Mediterranean marine fish based on a review of a database that provides 23 traits for 235 Mediterranean fish species. It has two major objectives: (1) analyze the distribution of traits in different fish species to identify rare traits and discuss their potential functional role, and (2) study the relationships between traits. The second objective would better defined if the potential confusion in the definition of traits is corrected.
Statistical methods are described in detail and are well-adapted to the data. If there is one weakness, it concerns the ANOVA test. It would be useful to indicate which tests/methods used to verify the normality and the homogeneity of the data prior to the analysis.

Validity of the findings

The discussion is generally based on a complete analysis data. It would benefit from being better organized by distinguishing functional traits from ecological traits and developing some superficially addressed themes. These themes are as follows:
• The proposal to use functional traits to promote an ecosystem-based analysis of traits by considering all biotic elements (plankton, nekton and benthos) of an ecosystem is relevant is certainly one of the major interests of using traits rather than species identity. It would be desirable to deepen this proposal by suggesting the common traits that could be taken into consideration in these different biotic elements.
• The authors suggested that the negative relationship between longevity and age at maturity which differs from previous study could be explained by the fact that age at maturity is expressed as the percentage of the life span and not in years. Using the raw data, this hypothesis should be tested.
• Different authors (e.g. Lavorel & Garnier, 2002; Violle et al., 2007) distinguish ‘response trait’ and ‘effect trait’. I would like the authors to question these notions in the choice of traits to be used to assess the effects of anthropogenic pressures or climate change.
• To make predictions for future scenarios, the authors argued that anthropogenic stressors may alter trait composition indirectly through the selection of associated traits (see Line 328). If I totally agree with this idea, some authors have also suggested that fishing may induce changes in some traits which are exposed to selective pressures and are heritable (see Edeline, E. et al. (2007) Trait changes in a harvested population are driven by a dynamic tug-of-war between natural and harvest selection. Proc. Natl. Acad. Sci. U. S. A. 104:15799–15804). I would appreciate that this issue on the heritability of some traits is discussed.
• Traits could be variable among individuals or among populations within a species (see Violle et al., 2017). How can you include this variability in the monitoring program? Do you consider that the trait should be measured from monitoring program only infer from species composition of fish communities?
• Line 224: Indicate that the reference to Verges et al. (2014) concerns the alien species Siganus recently introduced in the Mediterranean Sea.

Additional comments

The article provides a considerable amount of information on the biological characteristics of Mediterranean fish species. The analysis of the database established during this study provides useful information on the degree of rarity of certain traits and allows discussion of the functional role of rare traits in relation to population size. It also makes it possible to describe the relationships between traits. While this second part of the manuscript highlights expected relationships, it also shows the close links that can exist between functional traits and the environment. Such relationships can help to anticipate the consequences of climate change or anthropogenic pressures on functional diversity. An effort to define the traits and the distinction between functional and ecological traits would go a long way towards ensuring that the work done is in line with the proposed definitions of the traits. It would facilitate the organization of the manuscript and the selection of the main functional traits to be followed in an inventory management context. Some points of the discussion would also benefit from further elaboration.

·

Basic reporting

This study present an impressive database regarding the traits of more than 200 nektonic species from the Mediterranean sea. In addition to the gathering of such a comprehensive dataset that is extremely exciting for future macroecological research on the Mediterranean sea, the authors made a compelling effort to describe the trait distributions and trades-offs at a scale that has rarely been done before. Although the manuscript is well-written, adequately framed and is overall statistically valid, I believe that the major negative point of this study lie in the multiplicity of the tests performed and the lack of a real synthesis of this impressive amount of work (although some notable efforts have been done in this regard). Some of the statistical analyses performed, although mostly valid based on the type of data, results in many figures and tables and could easily be reframed into more synthetic and interpretable analyses. Although it remains perfectly understandable, in its current states the manuscript is hard to follow in some instances, and the key messages sometimes get lost in the amount of information given to the reader. Although I acknowledge the amount of work that some of my major comments may represent, I made several suggestion below that may help reach a more synthetic view of the data. Addressing these several points may I hope add value, understandability and clarity to this work and impressive database, which I feel is necessary in some parts, but in no way these comments should impede the publication of this work.

Experimental design

no comment

Validity of the findings

no comment

Additional comments
* * *
Major comments
* * *
------------1---------

The major drawback of this study is the consideration of only pairwise relationships, which results in many graphs and tests. Although I do not dismiss the value of these tests, they are hard to synthesize and do not considered the multicollinearity inherent to multivariate data. An easy way to summarise the groups of traits that covary and account for the multiple relationships existing among traits is to resort to multivariate analyses. Such multivariate analyses could be complemented by the existing tests, that are fully complementary to the approach I suggest below, but theses bivariate tests should only be for me an additional information, the core information being the description of the trait space through multivariate analyses.
Given the mixed type of the traits considered here, Hillsmith ordination (Hill and Smith, 1976) appears to be the most appropriate tool. The latter can be performed easily in R using the dudi.hillsmith function of the `ade4` package. See for e.g. Teichert et al. 2017

I think using such a multivariate approach could also broaden the scope of this manuscript in several ways. The authors have focused in several instances on the concept of rarity and such an approach would allow an easy identification of rare traits and of the species bearing them (the keystone species concept discussed in l. 63 and l.218 but that cannot be found anywhere in the results). In addition, it could allow to identify parts of the trait space with high redundancy, a concept that has been eluded in this manuscript and that should be more emphasized as it is as important as rare traits in terms of conservation and would allow a more comprehensive depiction of the Mediterranean nektonic trait space. See for e.g. Diaz et al. 2016 for a nice example on how to use multivariate approach to make a comprehensive description of trait spaces.

In addition, to go further on the idea of keystone species that I find particularly relevant and in the aim of reducing the complexity of the data and synthesizing information, functional groups could be delineated using clustering techniques as performed for example for Mediterranean copepods (Benedetti et al. 2016). It is fully complementary to the description of the trait space with ordination and the functional groups delineated could for example be represented on the ordination (see for e.g. Figure 4 of Benedetti et al. 2018).

Although the clustering of functional groups is more optional, I feel the inclusion of
an ordination (with something like the Figure 2 of Diaz et al. 2016) really necessary for synthesizing the results, simplifying the discussion regarding rare traits and traits trade-off, and developing more on the concept of keystone species and on the degree of redundancy of the Mediterranean nekton.

----------2------------

The choice of the 235 species should be more thoroughly discussed: why these 235 species and how representative is it of the diversity of nektonic species of the Mediterranean Sea? How much of the trait distribution you show here is actually relevant at the Mediterranean scale and how much of it is due to the species included and only described the database rather the Mediterranean nekton? On a similar note, does the database comprises of rare species? Is there commercial ones? I believe that answering these different questions with additional details on the pool of species included in the database would promote the reusing of this extensive dataset and would clarify the relevance of your results in terms of the ecology of the Mediterranean nekton.

---------3-------------

Although I feel several of the figures should be relegated to the Supplementary material with the inclusion of a multivariate approach (potentially Fig 3 and 4), I do have several comments on the understandability and readability of the figures and tables.
Fig 1 and 2 are fully relevant here and extremely interesting but I have some doubts about the representation chosen for continuous traits. Density distribution should be used instead of histograms and the threshold discussed in the text could then be highlighted through colors or shading (for e.g r - Shading a kernel density plot between two points. - Stack Overflow : https://stackoverflow.com/questions/3494593/shading-a-kernel-density-plot-between-two-points)

For Fig 1B, the x-axis should be named and it should be mentioned in the legend that individuals are ranked according to the mean of the range (if that is what is done). Also, there seems to be different shades of grey according to the bars. I am not sure what is the meaning of these.

I would prefer the panels of Fig 2 to be all on the same y-scale and to range from 0 to 100 to make the different dominance patterns clearer.

In Fig 3, in table 3 and in the text, the distinction made between linear and non-linear relationships are somehow obscured by the presence of log-transformed variables. I believe a clarification is needed here since a linear relationship between two log-transformed variables means cannot be called “a linear relationship between these two variables” as it is rather log-linear. Therefore, not only three pairs of variables show non-linear relationships (as said l. 185) but most of them do strictly speaking.

Also, if something like Table 3 and 4 was to be kept along with the multivariate analysis (for e.g. to support the significance of the relationships observed in the ordination), I would rather see one lm or glm per traits with all other traits (and potential polynomes of these traits) as explanatory variables and a table reporting the standardized slopes of each explanatory variables with the R2 and significance of the full model. This could allow to gather Table 3 and 4 into a single analysis where all variables (quantitative and qualitative) would be included. A selection of variables (through AIC for example; stepAIC function) would allow to reduce each model, and then the coefficients of the selected variables only would be reported into a single table reporting the best model for each traits (models that would contain potentially both continuous and categoric variables, as the coefficients of both are perfectly interpretable). See :

Schielzeth, H. (2010). Simple means to improve the interpretability of regression coefficients.Methods in Ecology and Evolution,1(2), 103-113.

Zuur, A. F., & Ieno, E. N. (2016). A protocol for conducting and presenting results of regression‐type analyses.Methods in Ecology and Evolution,7(6), 636-645.

Fig. 4 should be reorganized, either to group modalities belonging to the same trait together, or to group modalities with similar co-occurence patterns together.

---------4--------------

The choice of forcing each species to belong only to one category of a trait instead of allowing them to belong to more than one category using fuzzy coding, which allows accounting for some of the plasticity the species (Chevenet et al., 1994), should be at least discussed. Although fuzzy coding may be dependent upon expert knowledge when giving relative scores to the affinity of species to each modality of a trait, it seems a more realistic view of the plasticity of the species. The coding approach has a deep influence on the subsequent patterns described. Therefore, it should be made clear to the readers in the Material and Method section the type of coding that has been used and I believe that a quick discussion (that could potentially be a perspective of this work) on the matter would add value to the manuscript.
* * *
Minor comments
* * *
Thank you for providing the data, this will make a great contribution to future research. However, blank should preferentially be coded with “NA”. Also, if all data seem available, the range for “range type” traits are not provided (only the mean is), which impedes the reproducibility of Fig 1B and would also be an interesting information to make available in this database for future studies.

l.40. Functional traits “impact/influence” directly or indirectly the fitness of organisms but do not always increase it. Please change the “increase” accordingly to match with Violle’s definition of functional traits

Table 1 could be reorganized to put traits that describe similar features together (e.g. Traits 8/9 with traits 13/14/15; or trait 19 with trait 12; or traits 17/18 with trait 20).

l.47 to 64. The framing of the study is well done but I feel that mentioning references on key works that have described trait trades-off for other taxonomic groups and how they have allowed a more comprehensive and mechanistic understanding of these groups may make an even better case for this study (e.g. Litchman et al. 2013)

l.63. KeyStone instead of keynote

l.88. I would change “important” for “dominant” traits

l. 145-155 and l.236-241 : These results on the distribution of traits could be compared to other existing works to better put them in perspective. For e.g., although the categories included are not fully similar, I feel that discussing the similarities and differences in the proportion you find with those from the North Altantic and Pacific (See for e.g. Supplementary Figure 2 of Beukhof et al. 2019) may be worthwhile. Especially, herbivorous species are low while benthivorous species dominates in both databases. On the other hand, you did not include a “generalist” modality (which I think is a good thing) but this should be mentioned and discussed as it changes the relative proportion of other traits (especially piscivorous species apparently). Such comparison would allow to extend the generality of your findings and open-up towards some potential general trends regarding the structure of nekton trait space.

l.149-150 Can you also indicate the mean, median or mode of the distributions, as well as potentially the range of some variables when relevant

l.157 I am not sure of what you mean by “angle” and did not understand this sentence. Could you please clarify your thoughts ?

l.243 : weighting may be done using species abundance I agree, but may/should also be weighted by biomass (especially when looking at the role of traits on ecosystem rates and functioning)

l.253 Another way (and potentially more conclusive one) to address and confirm the non-random distribution of traits among species would be to assess whether or not these >200 nektonic species are distributed randomly in trait space using a randomization approach such as the ones performed in Diaz et al. 2016. However, I acknowledge that this may not fully fall within the scope of this study, but it may nonetheless be an interesting open-up/perspective.

l.298 In my opinion, many of the traits you included are rather « soft » (each traits are proxy of one or several functions rather than direct measures of these functions). Although the distinction between hard and soft is rather subjective, I feel many of these traits do not fit in the definition of hard traits given by Violle et al. 2007 (Oikos) or Nock et al. 2016 (Functional traits). Some of them such as temperature/depth range would better fall in the category of “ecological traits” as defined by Beauchard et al. 2017 (and see the potential limitations discussed in this paper for the use of these traits). Therefore, I would remove this term that is for me more than questionable here.

l.306-308. The distinction between effect and response traits may be relevant to formalise the distinction you are making here (see Suding et al. 2008)

l.326-329 : Suggested ref for the differential response to stress among fish functional groups and the need to understand traits to understand their responses : Henderson, C. J., Gilby, B. L., Schlacher, T. A., Connolly, R. M., Sheaves, M. , Maxwell, P. S., Flint, N. , Borland, H. P., Martin, T. S., Gorissen, B. and Olds, A. D. (2019), Landscape transformation alters functional diversity in coastal seascapes. Ecography. doi:[10.1111/ecog.04504](https://doi.org/10.1111/ecog.04504)

l.326-329 : Suggested ref for the need to incorporate trait and life-history knowledge of nektonic species to understand their dynamics : Marquez, J. F., Lee, A. M., Aanes, S. , Engen, S. , Herfindal, I. , Salthaug, A. and Sæther, B. (2019), Spatial scaling of population synchrony in marine fish depends on their life history. Ecol Lett, 22: 1787-1796. doi:10.1111/ele.13360

Table 5, penultimate line : temperature instead of temp/ture
* * *
Suggested ref
* * *
Beauchard, O., Veríssimo, H., Queirós, A. M., & Herman, P. M. J. (2017). The use of multiple biological traits in marine community ecology and its potential in ecological indicator development.Ecological Indicators,76, 81-96.

Fabio Benedetti, Stéphane Gasparini, Sakina-Dorothée Ayata, Identifying copepod functional groups from species functional traits,Journal of Plankton Research, Volume 38, Issue 1, January/February 2016, Pages 159–166,https://doi.org/10.1093/plankt/fbv096

Benedetti, F,Vogt, M,Righetti, D,Guilhaumon, F,Ayata, S‐D.Do functional groups of planktonic copepods differ in their ecological niches?.J Biogeogr.2018;45:604–616.https://doi.org/10.1111/jbi.13166

Chevenet, F., Dolédec, S., & Chessel, D. (1994). A fuzzy coding approach for the analysis of long‐term ecological data. Freshwater Biology, 31(3), 295–309. https://doi.org/10.1111/j.1365‐2427.1994.tb01742.x

Díaz, S., Kattge, J., Cornelissen, J. H., Wright, I. J., Lavorel, S., Dray, S., … & Garnier, E. (2016). The global spectrum of plant form and function._Nature_,_529_(7585), 167. The global spectrum of plant form and function | Nature

Hill, M. O. and Smith, A. J. (1976), PRINCIPAL COMPONENT ANALYSIS OF TAXONOMIC DATA WITH MULTI‐STATE DISCRETE CHARACTERS. TAXON, 25: 249-255. doi:10.2307/1219449

Litchman, E., Ohman, M. D., & Kiørboe, T. (2013). Trait-based approaches to zooplankton communities.Journal of Plankton Research,35(3), 473-484.

Nock, C. A., Vogt, R. J., & Beisner, B. E. (2001). Functional traits.eLS, 1-8.

SUDING, K. N., LAVOREL, S. , CHAPIN, F. S., CORNELISSEN, J. H., DÍAZ, S. , GARNIER, E. , GOLDBERG, D. , HOOPER, D. U., JACKSON, S. T. and NAVAS, M. (2008), Scaling environmental change through the community‐level: a trait‐based response‐and‐effect framework for plants. Global Change Biology, 14: 1125-1140. doi:10.1111/j.1365-2486.2008.01557.x

Teichert, N., Pasquaud, S., Borja, A., Chust, G., Uriarte, A., & Lepage, M. (2017). Living under stressful conditions: Fish life history strategies across environmental gradients in estuaries.Estuarine, Coastal and Shelf Science,188, 18-26. Redirecting

Supplementary material of Beukhof, Esther; Dencker, Tim S; Palomares, M L D;Maureaud, Aurore(2019):A trait collection of marine fish species from North Atlantic and Northeast Pacific continental shelf seas._PANGAEA_,(https://doi.org/10.1594/PANGAEA.900866)

---

## Round 0.2 · accepted · Accept

Your revisions are substantial and thorough, following the reviewers suggestions and comments.